# Heparin Provides Antiviral Activity Against Rhinovirus-16 via an Heparan Sulfate Proteoglycan-Independent Mechanism

**DOI:** 10.3390/ijms262110393

**Published:** 2025-10-25

**Authors:** Leanne C. Helgers, Killian E. Vlaming, Tanja M. Kaptein, Julia Eder, Jan Willem Duitman, Teunis B. H. Geijtenbeek

**Affiliations:** 1Department of Experimental Immunology, Amsterdam UMC Location University of Amsterdam, 1105 AZ Amsterdam, The Netherlands; k.e.vlaming@amsterdamumc.nl (K.E.V.); t.m.kaptein@amsterdamumc.nl (T.M.K.); jeder@cemm.oeaw.ac.at (J.E.); j.w.duitman@amsterdamumc.nl (J.W.D.); t.b.geijtenbeek@amsterdamumc.nl (T.B.H.G.); 2Amsterdam Institute for Infectious Diseases and Immunology, 1105 AZ Amsterdam, The Netherlands; 3Department of Pulmonary Medicine, Amsterdam UMC Location University of Amsterdam, 1105 AZ Amsterdam, The Netherlands

**Keywords:** human rhinovirus (HRV), heparan sulfate proteoglycans (HSPGs), heparin

## Abstract

Human rhinovirus 16 (HRV-16) is a major cause of common colds and can exacerbate asthma and COPD, yet no approved antiviral treatments exist. Heparin, a highly sulfated polysaccharide, is known to block viral infection of many viruses that require attachment to heparan sulfate proteoglycans (HSPGs). Here, we investigated whether heparin inhibits HRV-16 infection. HRV-16 uses ICAM-1 as its attachment receptor and lacks a confirmed HSPG-binding mechanism. Notably, heparin inhibited HRV-16 infection in vitro in a dose- and time-dependent manner. Pre-treatment of either cells or virus particles with unfractionated heparin significantly reduced HRV-16 RNA expression at 24 and 48 h post-infection. In contrast, low-molecular-weight heparins blocked infection of HRV-16 significantly less effectively compared to unfractionated heparins. Our findings suggest that the inhibitory effect of unfractionated heparin on HRV-16 infection is likely independent of specific HSPGs interactions and may be mediated by the size and highly negative charge of unfractionated heparin. Importantly, the ability of unfractionated heparin to block viruses that do not require HSPGs for attachment implies a broader antiviral potential as a prophylactic or therapeutic agent against a variety of respiratory viruses.

## 1. Introduction

Human rhinoviruses (HRVs) are single-stranded RNA viruses of the family *Picornaviridae* and the most common cause of upper respiratory tract infections [1,2]. There are over 150 HRV serotypes, divided into three species (A, B, C) based on sequence diversity [1,3,4,5]. Nearly all of the HRV-A and HRV-B serotypes, including HRV-16 (group A), utilize intercellular adhesion molecule-1 (ICAM-1) as their receptor for cell entry [1,6,7,8], with a minor portion utilizing the low-density lipoprotein (LDL) receptor family instead [9]. HRV has been identified as the most common cause of upper respiratory tract infection, primarily affecting children, the elderly, and those with pre-existing respiratory conditions. Infections with HRV are usually mild, but are also associated with severe bronchiolitis, community-acquired pneumonia, and exacerbations of asthma and chronic obstructive pulmonary disease (COPD). Hence, HRV16 poses serious risks in immunocompromised individuals [10,11]. There are no approved antiviral drugs for HRV infections, and current management is purely supportive, underscoring the need for new prophylactic and therapeutic strategies against HRVs.

Many different viruses initiate infection by attaching to cell-surface heparan sulfate proteoglycans (HSPGs) [12]. HSPGs are highly sulfated negatively charged transmembrane receptors that are expressed ubiquitously by different cell types. Unfractionated heparin is a large (~30 kDA) glycosaminoglycan structure that mimics the structure of heparan sulfate. Hence, unfractionated heparin can function as a soluble “decoy” receptor that blocks HSPG-dependent viruses from binding to cellular HSPGs. Indeed, unfractionated heparin has been shown to impede infections by HSPG-dependent viruses such as cytomegalovirus, dengue virus, enterovirus A71, hepatitis C virus, human Immunodeficiency Virus type 1, papillomavirus, and Severe Acute Respiratory Syndrome Coronavirus (SARS-CoV) 1 and 2 [12,13,14,15,16,17]. These prior findings establish a proof-of-principle that unfractionated heparin protects against a variety of viral infections. SARS-CoV-2, the coronavirus responsible for COVID-19, strongly depends on HSPGs to attach to cells, as an essential prerequisite to engage its ACE2 receptor for entry [18]. This dependency is mediated by a defined HSPG-binding site on the viral Spike (S) protein, which facilitates initial attachment to the host cell surface. Consistent with this, unfractionated heparin and low-molecular-weight heparins (LMWH) competitively block HSPGs and inhibit SARS-CoV-2 binding to epithelial cells [13,19,20]. This antiviral property of heparins has motivated investigations into inhaled or intranasal heparin as a prophylaxis against SARS-CoV-2. We previously showed that intranasal administration of enoxaparin significantly reduced SARS-CoV-2 attachment to human nasal epithelial cells in vivo with no adverse events [20]. Thus, heparins represent a promising class of host-directed antivirals for respiratory HSPG-dependent viruses.

While heparin’s antiviral effects are well documented for HSPG-dependent viruses, it is unclear whether heparin interferes with viruses that do not rely on HSPGs. However, heparin might also cause blockage due to its physical properties via altered electrostatic interactions or steric hindrance [21,22], suggesting an effect on HSPG-independent viruses. In this study, we investigated the protective effect of heparin against acute HRV-16 infection in vitro and demonstrated that unfractionated heparin inhibits HRV-16 infection. LMWHs suppressed HRV-16 infection to a limited extent compared to unfractionated heparin. Inhibition of infection by unfractionated heparin but not LMWHs suggests that unfractionated heparin inhibits HRV-16 via an HSPG-independent mechanism. These results indicate broader antiviral activity for unfractionated heparin beyond direct dependence on HSPG binding.

## 2. Results

### 2.1. Heparin Inhibits HRV-16 Infection in a Dose-Dependent Manner

To investigate the impact of heparin on HRV-16 infection, either H1-HeLa cells (cells) or HRV-16 virions (viruses) were pre-treated with heparin for 30 min prior to infection. Uninfected and untreated controls were included for comparison. Our study specifically aims to evaluate the acute antiviral effects of heparin (viral attachment and early infection). Hence, HRV-16 RNA was quantified by qPCR at 24 and 48 h post-infection (hpi). Heparin treatment resulted in a clear dose-dependent reduction in HRV-16 replication (Figure 1A,B). Reduction in viral replication was largely similar between the two conditions, with pre-treated H1-HeLa cells showing marginally better protection from infection compared to HRV-16 pre-treated with heparin. Inhibitory effects of heparin on HRV-16 expressions were observed at 250 IU/mL (24 hpi; Figure 1A; *p* < 0.0001) and 375 IU/mL (48 hpi; Figure 1B; *p* < 0.0001). At 24 hpi, there was an almost complete reduction observed at 750 IU/mL, while at 48 hpi, this was observed at 400 IU/mL (Figure 1). A significant increase in HRV-16 expression was observed at 48 hpi with a low heparin concentration (125 IU/mL). The unnormalized data are presented in Appendix A.

In parallel with viral RNA measurements, we assessed cell viability under similar conditions using the 3-(4,5-dimethylthiazol-2-yl)-2,5-diphenyltetrazolium bromide (MTT) assay. At 24 hpi, heparin had no significant effect on H1-HeLa cell viability relative to untreated controls (Figure 2A). However, at 48 hpi, mild cytotoxic effects were observed at the highest concentrations (750–1000 U/mL; Figure 2B). Importantly, the concentration of heparin (≤375 U/mL) at 48 hpi that achieved substantial antiviral effects showed little to no cytotoxicity (Figure 2B). Thus, heparin effectively blocks HRV-16 infection both at the viral and host sites. The unnormalized data are presented in Appendix A.

### 2.2. Unfractionated Heparin Is More Effective than LMWHs to Protect Against HRV-16

Given the strong inhibitory effect of unfractionated heparin on HRV-16, we next evaluated whether LMWHs could similarly protect against HRV-16 infection. We investigated the effect of three different LMWHs: enoxaparin, dalteparin, and tinzaparin, and compared them to unfractionated heparin at equivalent concentrations. As before, either H1-HeLa cells or HRV-16 was pre-treated with either heparin or a LMWH, and infection was determined at 24 and 48 hpi. At 24 hpi, unfractionated heparin (500 U/mL) showed a significant reduction in HRV-16 expression (Figure 3A). However, none of the LMWHs significantly reduced viral replication (Figure 3A), although there was a trend towards decreased HRV-16 infection for all three LMWHs. The replication block of HRV-16 by unfractionated heparin became more apparent at a higher concentration (1000 U/mL), while neither of the LMWHs showed a significant reduction in viral replication (Figure 3A). There were no observed differences between pre-treatment of H1-HeLa cells or HRV-16 virions. At 48 hpi, most LMWHs and conditions still did not significantly reduce viral infections (Figure 3B), with the exception of H1-HeLa pre-incubated with enoxaparin (*p* < 0.05) or tinzaparin (*p* < 0.05) at 1000 IU/mL (Figure 3B). These results indicate that while LMWHs can inhibit HRV-16 infection to a small extent, their inhibitory effect is much lower than that of unfractionated heparin. The unnormalized data are presented in Appendix A.

## 3. Discussion

In this study, we demonstrate that heparin has a protective effect against HRV-16 infection in vitro, despite HRV-16 not directly requiring HSPGs for cell entry. Unfractionated heparin inhibited HRV-16 in a dose- and time-dependent manner, while no significant cytotoxicity was observed for heparin at effective doses. In contrast, LMWHs were less effective in reducing HRV-16 infection. The results presented in this study show how heparin, but not LMWHs, blocks HSPG-independent viruses from causing infections, contributing to the known antiviral scope of heparin.

The ability of heparin to block viral infection has traditionally been linked to viruses that exploit HSPG during attachment [12,15,23]. HSPG-dependent viruses include coronaviruses, flaviviruses, and herpesviruses, and exogenous heparin competitively inhibits the virus-HSPG interaction, thereby preventing infections. SARS-CoV-2, for example, binds HSPGs as a co-receptor, and unfractionated heparin or LMWHs block viral attachment to HSPGs present on the cell surface [12,20]. Unfractionated heparin was previously shown to be substantially more potent than LMWHs against SARS-CoV-2 [24], likely because the longer heparin chains provide multiple binding sites for the viral spike, enhancing neutralization.

HRV-16 primarily uses ICAM-1 for entry and has no known dependence on HSPGs [1]. A prior study reported that heparin did not affect HRV-16 attachment in cell culture [25]. However, we observed a significant inhibitory effect of heparin on HRV-16 infectivity in vitro. One explanation may lie in the different experimental approaches: Bochkov et al. assessed virus attachment in the presence of heparin, whereas we evaluated the inhibitory effect of heparin on infection of HRV-16 [25]. Attachment alone does not guarantee successful infection, as virions may still attach but fail to internalize or fuse within endo/lysosomal compartments. Consistent with this, we observed inhibition of viral replication when either cells or virions were pre-incubated with heparin. In the latter case, heparin may adhere to positively charged regions of the HRV-16 capsid, partially masking the viral surface and impairing access to ICAM-1. Thus, even though HRV-16 does not depend on HSPGs, heparin could act as a “virus coating” agent, neutralizing virions through electrostatic interactions and preventing productive infection through other receptors.

The differential effects of unfractionated heparin and LMWHs offer insight into the mechanism of inhibition. Unfractionated heparin was markedly more effective against HRV-16, consistent with data showing that larger size and higher sulfation conferred stronger antiviral activity [24]. While LMWHs showed modest inhibition at high doses, they lacked the potency of unfractionated heparin, likely due to their smaller size (~4.5–5 kDa vs. 3–30 kDa) and fewer monosaccharide units (~13 vs. ~46), resulting in fewer viral binding sites [26,27,28]. LMWHs are derived from unfractionated heparin by controlled depolymerization, a process that breaks the long heparin chains into shorter, more uniform fragments. Despite their structural similarity, LMWHs are hypothesized to be less effective than unfractionated heparin at inhibiting HSPG-independent viruses. This is due to their smaller molecular size (~4–6 kDa), lower sulfation density, and reduced charge, which might limit their capacity for multivalent binding to viral surface proteins. While the greater antiviral potency of unfractionated heparin compared with LMWHs may relate to differences in size, sulfation density, and charge, these interpretations remain hypothetical. Further studies using enzymatic removal of HSPGs or comparisons with other polyanions would be required to directly confirm the mechanism.

Importantly, it should be noted that the use of anti-Xa units for normalizing heparin doses reflects anticoagulant activity against Factor Xa rather than the physical and chemical properties relevant for antiviral activity, such as molecular mass, sulfate content, and polysaccharide chain length. Consequently, 1000 IU of unfractionated heparin corresponds to a substantially greater mass and number of sulfate groups compared to 1000 IU of LMWHs. Thus, equating antiviral doses based solely on anti-Xa activity may be misleading. In our experiments, heparin demonstrated a clear and robust dose-dependent antiviral effect against HRV-16, an effect that was not observed with LMWHs within clinically relevant concentration ranges. These observations underscore the importance of considering molecular size, sulfation density, and overall structure when evaluating heparins for antiviral purposes, and highlight the limitations of direct comparisons based exclusively on anti-Xa units.

Encouragingly, heparin and LMWHs have shown favorable safety profiles when administered via the respiratory route. Clinical studies in COVID-19 patients have found that nebulized unfractionated heparin is safe and well tolerated, with no significant increase in bleeding complications or other serious adverse events [20,29]. In patients with acute lung injury, inhaled heparin has been associated with improved outcomes and faster recovery in some trials [30,31]. Our findings have potential implications for antiviral therapy. They suggest that heparin, and by extension other polyanionic drugs, could provide broad-spectrum protection against viruses like HRV-16 that do not directly rely on HSPG. This broad antiviral potential of heparin is encouraging, especially considering the ubiquity of rhinoviruses and the lack of existing treatments. An inhaled heparin-based intervention could be used prophylactically during the cold season or in high-risk populations to reduce the frequency and/or severity of HRV-induced exacerbations [1].

In conclusion, our study provides proof-of-principle that heparin inhibits HRV-16 infection in vitro and suggests that heparin and its derivatives could serve as broad-spectrum inhibitors of respiratory viruses. Further research is warranted to explore the use of intranasal or inhaled heparin in vivo for preventing or treating viral infections.

## 4. Materials and Methods

Cell Culture

H1-HeLa cells (ATCC CRL-1958) were cultured in Dulbecco’s modified Eagle’s high glucose medium (DMEM; Gibco Life Technologies, Carlsbad, CA, USA) supplemented with 10% fetal calf serum (FCS), l-glutamine (L-glut; 10 μg/mL), and penicillin/streptomycin (P/S; 10 μg/mL). All reagents were obtained from Sigma-Aldrich (Milwaukee, WI, USA). The cell culture was maintained at 33 °C with 5% CO_2_.

HRV-16 Production

Wild-type HRV-16 was obtained from Dr. Jan Willem Duitman, Department of Pulmonary Medicine, Amsterdam UMC. HRV-16 was cultured in H1-HeLa cells. H1-HeLa cells were inoculated with DMEM supplemented with 2% FCS, L-glut (10 μg/mL), and P/S (10 μg/mL) containing HRV-16 (TCID_50_ of 0.6 × 10^6^) and maintained at 33 °C with 5% CO_2_. After 2 h of pre-incubation, medium was added up to a total of 35 mL, and H1-HeLa cells were incubated for an additional 3–4 days. Supernatant containing HRV-16 was collected, centrifuged, and filtered using a 0.2 µm filter.

Viral titers were determined by 50% tissue culture infectious dose (TCID_50_) on H1-HeLa cells. In short, H1-HeLa cells were seeded in a 96-well plate at a cell density of 20,000 cells. The following day, the cells were inoculated with a 10-fold serial dilution of HRV-16 isolate in quadruplicate. Cell cytotoxicity was measured by assessing the cytopathic effect (CPE) using a microscope or by means of an MTT assay. Measurements were performed 3, 5, and 7 days after infection. Viral titer was determined as TCID_50_/mL and calculated based on the method first proposed by Reed and Muench [32].

HRV-16 infection

H1-HeLa cells are well-known to be susceptible to HRV-16 and support robust HRV-16 replication. H1-HeLa cells were seeded at a density of 20,000 cells in 100 µL in a 96-well plate. After 24 h, cells were exposed to the HRV-16 for 24–48 h. Additionally, H1-HeLa cells or HRV-16 was pre-incubated with a concentration range of unfractionated heparin (125 IU/mL–1000 U/ML) or LMWHs for 30 min prior to inoculation. For unfractionated heparin, 500 IU, and for LMWHs, 100 IU equals 1 mg. Infection was measured after 24–48 h at 33 °C with 5% CO_2_; cells were washed with PBS and lysed. Viral mRNA was determined by RT-qPCR.

Reverse Transcriptase-PCR

Total RNA was isolated using QIAamp Viral RNA Mini Kit (Qiagen, Hilden, Germany; cat # 52906) as per the manufacturer’s protocol. cDNA was synthesized with the M-MLV reverse transcriptase kit (Promega, Madison, WI, USA) and diluted 1 is to 5 before further application. PCR amplification was performed in a 7500 Fast Realtime PCR System (Applied Biosystems, Foster City, CA, USA). Primers and probes were designed with Primer Express 2.0 (Applied Biosystems, Foster City, CA, USA) and were obtained from Sigma-Aldrich. Primers used for GAPDH mRNA expression (RT-PCR): forward primer (CCATGTTCGTCATGGGTGTG), reverse primer (GGTGCTAA GCAGTTGGTGGTG). Amplification was measured in the presence of SYBR green as follows: 95 °C for 20 s, followed by 45 cycles at 95 °C for 3 s and 60 °C for 30 s. Melt curve stage included 95 °C for 15 s and 60 °C for 60 s, followed by 95 °C for 15 s and 60 °C for 15 s.

Primers used for HRV-16 mRNA expression (probe-based RT-PCR): forward primer (AGSCTGCGTGGCKGCC), reverse primer (ACACGGACACCCAAAGTAGT), and specific probe (6-FAM-TCCTCCGGCCCCTGAATGYGGCTAAYC-BHQ-1). HRV-16 amplification was measured using the GoTaq^®^ reaction mixture as follows: 50 °C for 2 min and denaturation at 95 °C for 10 min, followed by 45 cycles at 95 °C for 15 s and 60 °C for 60 s. PCR Probe Kits: GoTaq^®^ reaction mixture was obtained from Promega (Madison, WI, USA; cat # A6102).

Ct values were obtained for both household mRNA (GAPDH) and target mRNA (HRV-16). The expression of target genes was normalized to the household gene with the equation GAPDH (Nt = 2 Ct [GAPDH] − Ct [target]). Next, for relative HRV-16 expression, the condition of untreated (UT) HRV-16 infection was set at 1 for each donor.


**
*Tetrazolium dye colorimetric cell viability (MTT) assay*
**


MTT solution was added to H1-HeLa cells and incubated for 2–3 h at 37 °C. After removing the MTT solution, MTT solvent containing 4 mM HCl and 1% Nonidet P-40 (NP40) in isopropanol was added to the cells. Homogeneous solution was measured at optical density between 580 and 655 nm. Loss of MTT staining, as determined by a spectrometer (optical density at 580 nm [OD580]), was indicative of HRV-16-induced CPE.


**
*Statistics*
**


All results are presented as mean ± SD and were analyzed by GraphPad Prism 10.2.0 software (GraphPad Software Inc., La Jolla, CA, USA). A two-way ANOVA was performed for unpaired, non-parametric observations. Statistical significance was set at * *p* < 0.05, ** *p* < 0.01, *** *p* < 0.001, **** *p* < 0.0001.

## Figures and Tables

**Figure 1 ijms-26-10393-f001:**
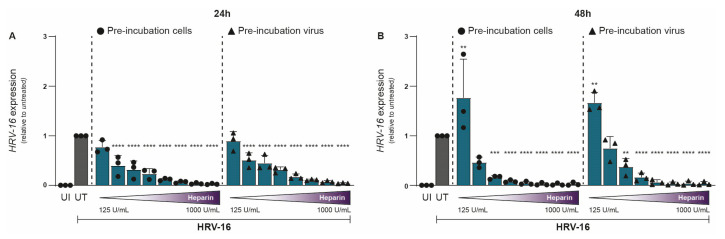
Antiviral effect of Heparin on HRV-16 infection in H1-HeLa cells. Either H1-HeLa cells (round) or HRV-16 particles (triangle) were pre-incubated with increasing concentration (125 IU/mL–1000I U/mL) of heparin (blue bar), and HRV-16 RNA expression was quantified at 24 (**A**) and 48 (**B**) hpi. using RT-PCR. Uninfected (UI; white bar) and HRV-16-infected but untreated (UT; gray bar) conditions were taken as controls. HRV-16 RNA expression is relative to the housekeeping gene GAPDH and was normalized for each experiment to the HRV-16 infected condition (**A**,**B**). The graphs represent collated data (mean ± SD) from 3 independent experiments using two replicates per experiment. A two-way ANOVA with Tukey’s multiple-comparison test was performed for unpaired observations. Statistical significance was set at ** *p* < 0.05, *** *p* < 0.005, **** *p* < 0.0005.

**Figure 2 ijms-26-10393-f002:**
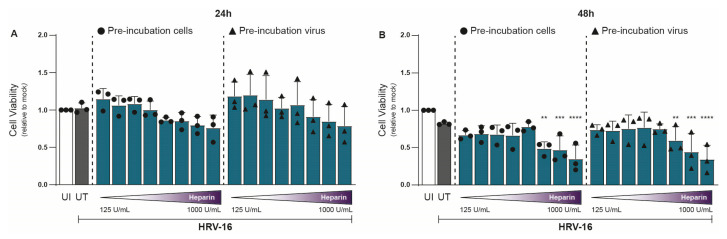
Heparin is tolerated by cells up to a level of 750 IU/mL at 48 hpi. Experimental setup and figure layout are similar to Figure 1. Cell viability was assessed using an MTT assay at 24 (**A**) and 48 (**B**) hpi, and loss of MTT staining was determined by a spectrometer (optical density at 580 nm [OD580]). Cell viability is indicated by OD580, and normalization of values to uninfected (UI) condition is shown (**A**,**B**). The graphs represent collated data (mean ± SD) from 3 independent experiments using two replicates per experiment. A two-way ANOVA with Tukey’s multiple-comparison test was performed for unpaired observations. Statistical significance was set at ** *p* < 0.05, *** *p* < 0.005, **** *p* < 0.0005.

**Figure 3 ijms-26-10393-f003:**
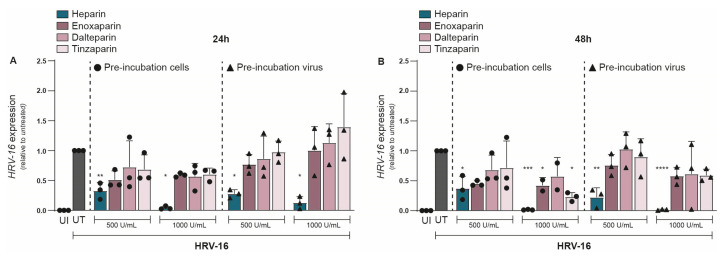
Antiviral effect of heparin but not LMWHs on HRV-16 replication. Either H1-HeLa cells (round) or HRV-16 particles (triangle) were pre-incubated with two different concentrations (500 IU/mL and 1000 IU/mL) of heparin (blue bar), or LMWHs enoxaparin (dark pink), dalteparin (medium pink), or tanzaparin (light pink). Uninfected (UI; white bar) and HRV-16-infected but untreated (UT; gray bar) conditions were taken as controls. HRV-16 RNA expression was quantified at 24 (**A**) and 48 (**B**) hpi. using RT-PCR. HRV-16 expression is relative to the housekeeping gene GAPDH, and normalization of values to UT reflects the relative reduction in infection (**A**,**B**). The graphs represent collated data (mean ± SD) from 3 independent experiments using two replicates per experiment. A two-way ANOVA with Tukey’s multiple-comparison test was performed for unpaired observations. Statistical significance was set at * *p* < 0.05, ** *p* < 0.05, *** *p* < 0.005, **** *p* < 0.0005.

## Data Availability

The authors confirm that the data supporting the findings of this study are available within the article and/or its Appendix A. Raw data were generated at Amsterdam UMC–location AMC, department of experimental immunology. Derived data supporting the findings of this study are available from the corresponding author, L.C. Helgers will be provided on request.

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
