# Peer review of "Heparin Provides Antiviral Activity Against Rhinovirus-16 via an Heparan Sulfate Proteoglycan-Independent Mechanism"

_ijms, 2025, doi:10.3390/ijms262110393_

Round 1
Reviewer 1 Report
Comments and Suggestions for Authors
The manuscript explores the inhibitory effect of unfractionated heparin against human rhinovirus 16 in vitro, proposing an HSPG-independent mechanism. The study addresses a timely and relevant topic, given the lack of antivirals for these viruses and the ongoing interest in host-directed antivirals such as heparins. The results are potentially important, suggesting a broader antiviral action of unfractionated heparin. However, the manuscript requires substantial revisions before it can be considered for publication. The concerns relate to clarity of writing, methodological rigor, justification of experimental design, and interpretation of results.
Minor Comments
Line 19 – The sentence “Notably, heparin exhibited HRV-16 infection in vitro in a dose- and time-dependent manner” is incorrect. It should read “heparin inhibited/reduced HRV-16 infection”. As written, the meaning is reversed.
Line 113 – “heparin nor a fractured anticoagulant” should read “heparin or a LMWH”.
Line 116 – P > 0.005 is given alongside a claim of significant reduction. Please clarify as this appears inconsistent.
Line 129 – Concentrations are reported as U/ML. Please standardize U/mL.
Line 132 – “rela-tive” should be corrected to “relative”.
Line 220 – The filter is described as “0.2 nm”, which is physically impossible. It should be 0.2 µm.
Figure Legends should specify the number of biological replicates, the control used for normalization (UT or UI), and the statistical tests applied.
Several terms are misspelled: “Firgure”, “enoxaparine” and “tinzaparin”.
“Household gene” should be corrected to “housekeeping gene”.
ICAM-1 is expanded as “Intracellular adhesion molecule-1” in the Abbreviations section. It should be “Intercellular adhesion molecule-1”.
The introduction would benefit from greater emphasis on the epidemiological burden of HRV infections, their role in asthma and COPD exacerbations, and the lack of specific antiviral therapies. This would strengthen the rationale for the study.
Major Comments
The chosen concentrations (125 – 1,000 IU/mL) are not sufficiently justified. Previous reports using heparin against other viruses (e.g., Zika, SARS-CoV-2, Nipah) typically range from 4-250 µg/mL. The doses here are much higher than those commonly tested. A rationale is needed for these doses and preferably report concentrations in µg/mL or mg/mL, since international units reflect anticoagulant activity rather than physicochemical properties relevant to antiviral activity.
Only two time points were examined. This limited kinetic analysis cannot adequately represent the viral replication cycle or the dynamics of inhibition. A more detailed time-course would clarify whether heparin acts on viral entry, replication, or later stages.
Cytotoxicity was assessed exclusively by MTT assays at 24 h and 48 h. MTT alone might be insufficient, since heparin can alter metabolic activity without directly reflecting cell viability. Additional assays (LDH release, Annexin V/PI staining) and extended time points (72-96 h) would strengthen the claim that the antiviral effects are not secondary to cytotoxicity.
The discussion suggests that unfractionated heparin is more active than LMWHs due to its larger size, higher sulfation density and greater negative charge. While plausible, these interpretations remain speculative. Enzymatic removal of HSPGs or comparisons with other polyanions would be required to support the proposed mechanism.
The Methods section is unclear about whether SYBR Green or hydrolysis probes were used for each target. It should be explicitly stated which was applied to GAPDH and which to HRV-16. The statement that values were “set to 1 for each donor” is confusing in the context of HeLa cell culture.
The manuscript reports that a “two-way ANOVA for unpaired, non-parametric observations” was used. This is contradictory, as ANOVA assumes normality. If data are non-parametric, Kruskal-Wallis with appropriate post-hoc tests, should be used. Please also specify the number of biological vs technical replicates and describe corrections for multiple comparisons.
Author Response
We thank the Editor and Reviewers for their careful evaluation of our manuscript. We have addressed each point in detail below. All modifications are reflected in the revised manuscript, with new or modified text highlighted in the manuscript.
Reviewer 1
Comments and Suggestions for Authors
The manuscript explores the inhibitory effect of unfractionated heparin against human rhinovirus 16 in vitro, proposing an HSPG-independent mechanism. The study addresses a timely and relevant topic, given the lack of antivirals for these viruses and the ongoing interest in host-directed antivirals such as heparins. The results are potentially important, suggesting a broader antiviral action of unfractionated heparin. However, the manuscript requires substantial revisions before it can be considered for publication. The concerns relate to clarity of writing, methodological rigor, justification of experimental design, and interpretation of results.
Minor Comments
We thank the reviewer for the comments below and we apologize for the mistakes and we have addressed these comments in the manuscript.
- Line 18: The sentence “Notably, heparin exhibited HRV-16 infection in vitro in a dose- and time-dependent manner” is incorrect. It should read “heparin inhibited/reduced HRV-16 infection”. As written, the meaning is reversed.
- Line 113 : “heparin nor a fractured anticoagulant” should read “heparin or a LMWH”.
- Line 115: P > 0.05 is given alongside a claim of significant reduction. Please clarify as this appears inconsistent.
- Line 129: Concentrations are reported as U/ML. Please standardize U/mL.
- Line 132: “rela-tive” should be corrected to “relative”.
- Line 220: The filter is described as “0.2 nm”, which is physically impossible. It should be 0.2 µm.
- Line 115 - 132: Several terms are misspelled: “Firgure”, “enoxaparine” and “tinzaparine”.
- Abbreviation table: ICAM-1 is expanded as “Intracellular adhesion molecule-1” in the Abbreviations section. It should be “Intercellular adhesion molecule-1”.
- Figure Legends should specify the number of biological replicates, the control used for normalization (UT or UI), and the statistical tests applied.
- “Household gene” should be corrected to “housekeeping gene”.
- The introduction would benefit from greater emphasis on the epidemiological burden of HRV infections, their role in asthma and COPD exacerbations, and the lack of specific antiviral therapies. This would strengthen the rationale for the study.
Major Comments
1). The chosen concentrations (125 – 1,000 IU/mL) are not sufficiently justified. Previous reports using heparin against other viruses (e.g., Zika, SARS-CoV-2, Nipah) typically range from 4-250 µg/mL. The doses here are much higher than those commonly tested. A rationale is needed for these doses and preferably report concentrations in µg/mL or mg/mL, since international units reflect anticoagulant activity rather than physicochemical properties relevant to antiviral activity.
The concentrations used (125–1,000 IU/mL) were chosen based on preliminary experiments showing antiviral activity without cytotoxicity at 24–48 h. While IU reflect anticoagulant activity, we used IU/mL here to align with clinically relevant formulations because our goal is nasal, surface-level inhibition of viral binding. For unfractionated heparin 500 IU and for LMWHs 100 IU equals 1 mg. We have clarified the concentration in mg in the methods section of the manuscript (line: 259-260).
- Only two time points were examined. This limited kinetic analysis cannot adequately represent the viral replication cycle or the dynamics of inhibition. A more detailed time-course would clarify whether heparin acts on viral entry, replication, or later stages.
Our study specifically aims to evaluate the acute antiviral effects of heparin (viral attachment and early infection), rather than later stages of the viral replication cycle. Our goal is to deliver heparin using a nasal spray and due to mucociliary clearance heparin should not be present for an extended period of time (>24h). Accordingly, we focused on the early phase of infection and assessed viral inhibition at 24 and 48 hours, which captures the relevant window for assessing the antiviral effects of heparin on HRV-16. A more extended kinetic analysis would primarily reflect downstream replication events, which are not the target of our intervention. Therefore, the chosen time points are sufficient to demonstrate the mechanism of action relevant to our prophylactic approach. We have clarified this in the manuscript (line: 71; 81-83).
- Cytotoxicity was assessed exclusively by MTT assays at 24 h and 48 h. MTT alone might be insufficient, since heparin can alter metabolic activity without directly reflecting cell viability. Additional assays (LDH release, Annexin V/PI staining) and extended time points (72-96 h) would strengthen the claim that the antiviral effects are not secondary to cytotoxicity.
We thank the reviewer for this suggestion. In our study, we focused on the acute antiviral effects of heparin against HRV-16 which occur within the first 24–48 hours post-infection. Hence, the relevant antiviral window is limited to the early phase of viral attachment and infection. Assessing cytotoxicity at 24 and 48 hours using MTT is relevant to ensure that the observed antiviral effects are not secondary to acute cytotoxicity. Moreover, our data show that heparin hardly affects cytotoxicity even at 48 hours post infection, which support our data that the observed block is not due to cytotoxicity. We have clarified this in the manuscript (line: 71; 81-83).
- The discussion suggests that unfractionated heparin is more active than LMWHs due to its larger size, higher sulfation density and greater negative charge. While plausible, these interpretations remain speculative. Enzymatic removal of HSPGs or comparisons with other polyanions would be required to support the proposed mechanism.
We appreciate the reviewer’s comment and agree that additional experiments, such as enzymatic removal of HSPGs or comparative testing with other polyanionic compounds, would further strengthen the mechanistic understanding. However, these experiments are beyond the scope of this study. We have now clarified this limitation in the revised Discussion section (line: 194-197):
- The Methods section is unclear about whether SYBR Green or hydrolysis probes were used for each target. It should be explicitly stated which was applied to GAPDH and which to HRV-16. The statement that values were “set to 1 for each donor” is confusing in the context of HeLa cell culture.
Thank you for this observation. We have clarified the methods sections accordingly (line: 268-282).
- The manuscript reports that a “two-way ANOVA for unpaired, non-parametric observations” was used. This is contradictory, as ANOVA assumes normality. If data are non-parametric, Kruskal-Wallis with appropriate post-hoc tests, should be used. Please also specify the number of biological vs technical replicates and describe corrections for multiple comparisons.
We recognize our mistake that the reviewer highlighted and have included the correct test in all figure legends. In addition, we included the number of biological and technical replicates in the figure legends as well.

Reviewer 2 Report
Comments and Suggestions for Authors
The present study "Heparin Provides Antiviral Activity Against Rhinovirus-16 via an Heparan Sulfate Proteoglycan-Independent Mechanism" is focused on the antiviral effect of heparin in inhibiting HRV-16 infection; in contrast, low molecular weight heparins blocked infection of HRV-16 significantly less effectively compared to unfractionated heparin. The article's aims are relevant, but they are not fully expressed throughout the text.
Below are some suggestions for improving the quality and clarity of the study:
- Figures are confusing. Authors should show data in a clearer and readable manner.
- Why did the authors use H1-HeLa cells?
- The symbol * should be replaced by x.
- Which kind of primers did the author mean for HRV-16?
- Lines 76–77: I recommend removing the words in parentheses, “cells” and “viruses,” as they are redundant.
- Include the p-values in the figure legends instead of the main text to improve readability.
- Lines 91 and 132: I recommend replacing the term “RT-PCR” with “qPCR”, as the former may be confused with reverse transcription PCR.
- It would be beneficial to create a separate paragraph dedicated to cytotoxicity to provide a clearer and more structured overview.
- Lines 111 and 123: I would suggest that the authors maintain coherent use of the compound names (enoxaparin, tinzaparin, dalteparin) throughout the paragraph.
- Lines 113,116, and 132: Correct the typographical errors, such as “Firgure” should be “Figure,” “nor a fractured anticoagulant” should be revised to “or a fractionated anticoagulant,” and “rela-tive” should be corrected to “relative.”
- Lines 156-161: The phrase "Attachment does not necessarily equate to successful infection" could be revised using more accurate scientific language, such as "Attachment alone does not guarantee successful infection”. Additionally, the verb "adhere to" is not ideal in a molecular context, replacing it with "bind to".
- Also, the phlebovirus SFNV requires heparan sulphate to enter host cells. Please see the study: Chianese A, et al. Melittin-Related Peptides Interfere with Sandfly Fever Naples Virus Infection by Interacting with Heparan Sulphate. Microorganisms. 2023;11(10):2446. doi: 10.3390/microorganisms11102446.
Author Response
We thank the Editor and Reviewers for their careful evaluation of our manuscript. We have addressed each point in detail below. All modifications are reflected in the revised manuscript, with new or modified text highlighted in the manuscript.
Rewiever 2
Comments and Suggestions for Authors
The present study "Heparin Provides Antiviral Activity Against Rhinovirus-16 via an Heparan Sulfate Proteoglycan-Independent Mechanism" is focused on the antiviral effect of heparin in inhibiting HRV-16 infection; in contrast, low molecular weight heparins blocked infection of HRV-16 significantly less effectively compared to unfractionated heparin. The article's aims are relevant, but they are not fully expressed throughout the text.
Minor concerns:
We thank the reviewer and we have addressed all minor comments and highlighted this throughout the revised manuscript.
- Lines 76–77: I recommend removing the words in parentheses, “cells” and “viruses,” as they are redundant.
- Lines 91 and 132: I recommend replacing the term “RT-PCR” with “qPCR”, as the former may be confused with reverse transcription PCR.
- Lines 111 and 123: I would suggest that the authors maintain coherent use of the compound names (enoxaparin, tinzaparin, dalteparin) throughout the paragraph.
- Lines 113,116, and 132: Correct the typographical errors, such as “Firgure” should be “Figure,” “nor a fractured anticoagulant” should be revised to “or a fractionated anticoagulant,” and “rela-tive” should be corrected to “relative.”
- Lines 156-161: The phrase "Attachment does not necessarily equate to successful infection" could be revised using more accurate scientific language, such as "Attachment alone does not guarantee successful infection”. Additionally, the verb "adhere to" is not ideal in a molecular context, replacing it with "bind to".
- The symbol * should be replaced by x.
- Include the p-values in the figure legends instead of the main text to improve readability.
- It would be beneficial to create a separate paragraph dedicated to cytotoxicity to provide a clearer and more structured overview.
- Also, the phlebovirus SFNV requires heparan sulphate to enter host cells. Please see the study: Chianese A, et al. Melittin-Related Peptides Interfere with Sandfly Fever Naples Virus Infection by Interacting with Heparan Sulphate. Microorganisms. 2023;11(10):2446. doi: 10.3390/microorganisms11102446.
Major concerns
Figures are confusing. Authors should show data in a clearer and readable manner.
We have adjusted the figures and hope the data presentation is more clear.
Why did the authors use H1-HeLa cells?
H1-HeLa cells are very susceptible to HRV-16 and support robust HRV-16 replication. H1-HeLa cells are used in multiple HRV-16 studies, facilitating comparison with prior literature and ensuring consistency of viral replication. We have clarified this in the manuscript (line: 255-256).
Which kind of primers did the author mean for HRV-16?
We thank the reviewer for this comment. We have used HRV-16, forward primer (AGSCTGCGTGGCKGCC), reverse primer (ACACGGACACCCAAAGTAGT), and a specific probe (6-FAM-TCCTCCGGCCCCTGAATGYGGCTAAYC-BHQ-1). We have now further clarified this in the methods section of the manuscript (line: 268-282).

Round 2
Reviewer 1 Report
Comments and Suggestions for Authors
The revised version of the manuscript demonstrates a clear improvement in both structure and scientific clarity.
-All typographical errors and inconsistencies in units, gene nomenclature, and abbreviations have been correccted. The figures and legends provide the required information regarding replicates and statistical tests.
-The text reads more fluidly and the arguments are presented in a logical and coherent mantter. The introduction has been strengthened by including relevant context and highlighting the clinical significance of HRV infections.
-The Methods section has been significantly improved.
-The Discussion section integrates the rationale behind heparin's antiviral protential more effectively.
-The conclusions are now better aligned with the data presented.
Reviewer 2 Report
Comments and Suggestions for Authors
The article is now suitable for acceptance.